# Enhanced recovery following hip and knee arthroplasty: a systematic review of cost-effectiveness evidence

Mark G Pritchard ![ORCID],[1,2] Jacqueline Murphy ![ORCID],[1,3] Lok Cheng,[1,4] Roshni Janarthanan,[1,2] Andrew Judge ![ORCID],[5,6] Jose Leal ![ORCID] [1]

MGP and JM contributed equally.

For numbered affiliations see end of article.

**Correspondence to**
Dr Jose Leal;
jose.leal@dph.ox.ac.uk

## Abstract

**Objectives** To assess cost-effectiveness of enhanced recovery pathways following total hip and knee arthroplasties. Secondary objectives were to report on quality of studies and identify research gaps for future work.

**Design** Systematic review of cost–utility analyses.

**Data sources** Ovid MEDLINE, Embase, the National Health Service Economic Evaluations Database and EconLit, January 2000 to August 2019.

**Eligibility criteria** English-language peer-reviewed cost–utility analyses of enhanced recovery pathways, or components of one, compared with usual care, in patients having total hip or knee arthroplasties for osteoarthritis.

**Data extraction and synthesis** Data extracted by three reviewers with disagreements resolved by a fourth. Study quality assessed using the Consensus on Health Economic Criteria list, the International Society for Pharmacoeconomics and Outcomes Research and Assessment of the Validation Status of Health-Economic decision models tools; for trial-based studies the Cochrane Collaboration's tool to assess risk of bias. No quantitative synthesis was undertaken.

**Results** We identified 17 studies: five trial-based and 12 model-based studies. Two analyses evaluated entire enhanced recovery pathways and reported them to be cost-effective compared with usual care. Ten pathway components were more effective and cost-saving compared with usual care, three were cost-effective, and two were not cost-effective. We had concerns around risk of bias for all included studies, particularly regarding the short time horizon of the trials and lack of reporting of model validation.

**Conclusions** Consistent results supported enhanced recovery pathways as a whole, prophylactic systemic antibiotics, antibiotic-impregnated cement and conventional ventilation for infection prevention. No other interventions were subject of more than one study. We found ample scope for future cost-effectiveness studies, particularly analyses of entire recovery pathways and comparison of incremental changes within pathways. A key limitation is that standard practices have changed over the period covered by the included studies.

**PROSPERO registration number** CRD42017059473.

## Strengths and limitations of this study

- ► This systematic review of enhanced recovery pathways for hip and knee arthroplasties had a detailed search strategy, including entire pathways and their components.
- ► Appropriate tools were used to assess quality and validity of models, trials and economic evaluations.
- ► Conclusions were reliant on the availability, quality and validity of published studies into cost-effectiveness of hip and knee arthroplasty.

performed in 2011 within Organisation for Economic Co-operation and Development countries,[1 2] and rates continue to increase.[3–6] Enhanced recovery is a multimodal approach to reduce surgical morbidity and mortality. Recognising that factors other than surgical technique affect patient outcomes, Kehlet[7] considered how to optimise the preoperative, intraoperative and postoperative phases of patient care. These principles have been further developed specifically within the context of hip and knee arthroplasty.[8–15] Common components of an enhanced recovery pathway for hip and knee arthroplasty are listed in box 1. A recent systematic review[16] found that enhanced recovery after hip and knee arthroplasties reduced length of stay in hospital. The authors cited a study from New Zealand[17] which found an enhanced recovery pathway to be cost saving, but the study did not include any data on cost-effectiveness.

Cost–utility analyses have become the preferred approach to inform decisions on healthcare resource allocation.[18 19] In these, the effects of treatments are measured in quality-adjusted life years (QALY): the product of health-related quality of life (anchored at 0 for death and 1 for perfect health), and the time (in years) spent experiencing that level of health. Incremental cost-effectiveness ratios (ICER) are used to

## INTRODUCTION

Hip and knee arthroplasties are common procedures: around 1 million of each were

Box 1   Suggested components of an enhanced recovery pathway for hip and knee arthroplasty patients, data from references cited in text.

**Preoperative**
- ► Education.
- ► Discharge planning.
- ► Multidisciplinary assessment.
- ► Neuromuscular electrical stimulation.
- ► Nutrition screening.
- ► Optimisation of comorbidities.
- ► Physiotherapy.
- ► Premedication (possibly including standardised analgesia and steroids).
- ► Pulsed electromagnetic fields.

**Intraoperative**
- ► Standardised intravenous fluids.
- ► Avoid unnecessary blood transfusion.
- ► Minimally invasive surgery.
- ► Reduce heat loss.
- ► Specified anaesthetic requirements, such as spinal anaesthesia.
- ► Local infiltration of anaesthesia.
- ► Tranexamic acid.
- ► Prophylactic antibiotics.
- ► Intravenous dexamethasone.
- ► Avoid unnecessary drains.
- ► Computer-assisted surgery.

**Postoperative**
- ► Analgesia.
  - – Continuous neural block.
  - – Standardised multimodal analgesia, scheduled and as required.
  - – Reduce opioid use/avoid patient-controlled intravenous opioid analgesia.
- ► Scheduled antiemetic.
- ► Physical therapy starting on day of surgery.
- ► Avoid unnecessary blood transfusion.
- ► Oxygen administration.
- ► Avoid sleep disturbances.
- ► Early oral nutrition.
- ► Wound care.
- ► Thromboprophylaxis.
- ► Neuromuscular electrical stimulation.
- ► Aim for early discharge.

compare a treatment to a less effective alternative and a threshold value is used to determine whether it is cost-effective. Different countries have different thresholds for how much they are willing to pay per QALY gained.[20]

Systematic reviews of cost-effectiveness analyses have considered arthroplasty versus conservative management,[21 22] and specific components of enhanced recovery such as thromboprophylaxis.[23–25] However, we are not aware of any systematic reviews investigating the cost-effectiveness of a complete enhanced recovery pathway, or of most of the components. Our aim was to assess the cost-effectiveness evidence of enhanced recovery for patients having hip and knee arthroplasty. Specifically, we were interested in studies of adults having total hip arthroplasty (THA) or total knee arthroplasty (TKA) for

osteoarthritis, comparing an enhanced recovery pathway or components of one against usual care. Secondarily, we wanted to report study quality and identify research gaps for future work.

## METHODS

The complete methods are available in the published protocol,[26] registered with the International Prospective Register of Systematic Reviews, number CRD42017059473.[27] The selection of electronic databases and the search strategy were developed with an information specialist and in line with the preferred reporting items for systematic reviews and meta-analyses.[28] We searched Ovid MEDLINE, Embase, the National Health Service Economic Evaluations Database (via the Cochrane Library) and EconLit (via ProQuest) for English-language peer-reviewed papers published between 1 January 2000 and 1 March 2017 which included a cost–utility analysis of an enhanced recovery pathway or components of one, compared with usual care in patients having hip or knee arthroplasties (the complete search strategy for each database is presented in online supplementary table A1). Additional publications meeting our inclusion criteria were identified from the reference lists of the included studies. We repeated our search in August 2019 to identify more recently published studies. Our target population was adults having surgery for osteoarthritis. Studies exclusively concerning populations with other indications for surgery were excluded. We included studies with patients having different indications for surgery if the majority had osteoarthritis, or if the presented results allowed independent extraction of data for the subpopulation with osteoarthritis. Given that osteoarthritis is an indication for 92% of hip and 96% of knee arthroplasties,[29] we assumed that studies not providing details of the indication for surgery were representative of a population with osteoarthritis and therefore included these studies. Evaluations of surgical technique or choice of implant were excluded.

Studies were independently screened based on their titles and abstracts by four reviewers (MGP, JM, LC and JL). Full texts were obtained for studies chosen for inclusion by any reviewer. As an amendment to the protocol, evaluations of thromboprophylaxis were excluded at the full-text stage due to a recent comprehensive systematic review in that area.[23] Data extraction was performed for remaining studies by three reviewers (MGP, JM and RJ), with disagreements resolved by a fourth reviewer (JL). The data extraction proforma is included in the published protocol.[26]

To assess the quality of studies, we completed the Consensus on Health Economic Criteria (CHEC) list[30] for each publication. For model-based studies, we also used the questionnaire produced by the International Society for Pharmacoeconomics and Outcomes Research (ISPOR),[31] and the Assessment of the Validation Status of Health-Economic (AdViSHE) decision models tool.[32] We

added the Cochrane Collaboration's tool[33] to the original protocol to assess risk of bias in trial-based studies, referring to the original reports of trial outcomes where necessary. We assessed the quality of the data sources used in the studies according to a prespecified hierarchy of evidence tool (see online supplementary table A2).[34 35] For each component of a study, a score of 1 represented the use of the most appropriate data source, with increasing numbers representing progressively less appropriate sources.

The principal outcomes were a point estimate of cost-effectiveness in terms of incremental cost per QALY gained, and the probability of an intervention being cost-effective according to the willingness to pay threshold used by the authors of each study. We also examined whether an intervention could be cost-effective in a different country or setting than the original study. Hence, we assumed the studies to be generalisable and the ICERs to be comparable after conversion into 2016 purchasing power parity international dollars (US PPP).[36] As the studies had different methodologies and looked at

different combinations of components of the enhanced recovery pathway, no quantitative synthesis of the study results was attempted.

### Patient and public involvement
No patient involved.

## RESULTS
### Study selection
Our original search identified 11 060 publications, and one additional publication was found from other sources. We repeated the search in August 2019 and identified 3903 additional studies. After excluding duplicates, we screened 8657 titles and abstracts. We excluded 8482 papers based on their abstracts. For one study, we were unable to obtain the full text but it did not appear to be a cost-effectiveness study from the abstract. We therefore reviewed 174 full texts. We excluded 157 studies following

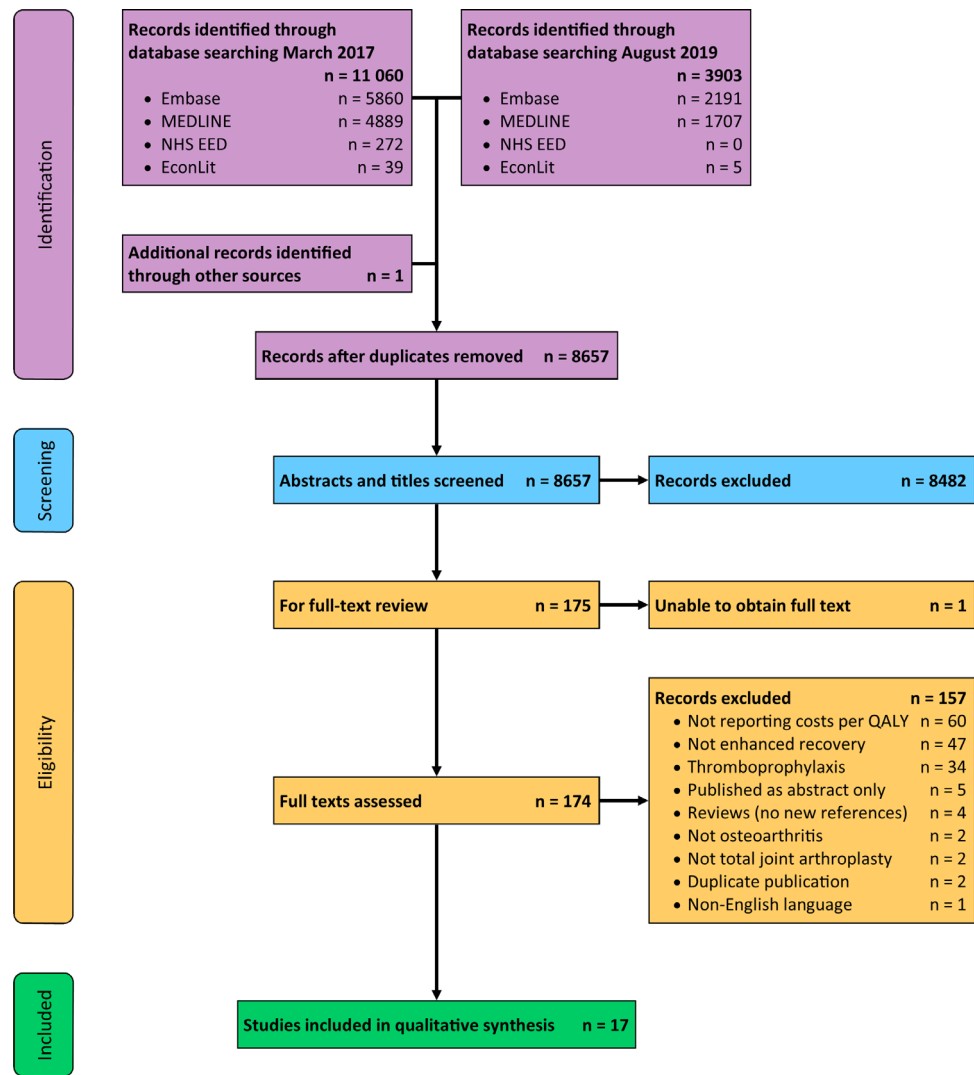

**Figure 1** Flow diagram of studies included in this review and reasons for exclusion, modified from Moher et al.[28] NHS EED, National Health Service Economic Evaluations Database; QALY, quality-adjusted life year.

review of their full texts (reasons for exclusion given in figure 1) and included 17 papers in this review.[37–53]

## Overview of included studies

Table 1 summarises the 17 included studies. Eight studies included both THA and TKA,[37 38 40–43 45 49] five only THA[44 46–48 53] and four only TKA.[39 50–52] Two papers looked at an entire enhanced recovery pathway.[37 38] We identified cost–utility evaluations of optimisation of comorbidities (specifically morbid obesity),[39] measures to reduce allogenic blood transfusion,[42–44] local infiltration of anaesthetic,[45] prophylactic antibiotics and other infection prevention measures[41 46–49] and physical therapy before[40] or after surgery.[40 50–52] The final study[53] concerned the optimal timing of follow-up which, although not included in our prior list of enhanced recovery interventions, we have included here as being allied to the pathway. Five studies were trial-based economic evaluations,[37 38 40 45 51] the remaining 12 being model-based. Eight studies were from Europe,[37 38 40 45 47 49–51] seven from the USA[39 41–44 46 52] and two from Australia.[48 53] Perspective, time horizon, discount rates and price year used in each study are reported in online supplementary table A3. The study perspective varied from only hospital costs[42–44 46 48 52 53] to a broader (societal) cost perspective.[37–41 49 50] Types of costs captured in each study are reported in online supplementary table A4. Of the seven studies eliciting utilities, QALYs were informed by utility values based on the EQ-5D-3L instrument in six studies[37 40 45 50 52 54] and on the 15D instrument in one study (online supplementary table A5).[51]

## Assessment of study and reporting quality

Using the CHEC list, the quality of the studies was generally good (figure 2A). Items raising more concerns were the short time horizons of the five trial-based studies (1 year)[37 38 40 45 51] and three model-based studies (between 6 weeks[49] and 7 years[53]). Using the Cochrane Collaboration's tool (figure 2B) for the five trials[37 54–57] studies were based on, the risk of bias was low for items such as incomplete outcome data and selective reporting but high or uncertain for the remainder. Four trial reports stated that the participants were allocated at random,[54–57] and the fifth was a before and after trial whereby patients were recruited consecutively from a waiting list.[37] Larsen and colleagues[54] discussed stratification, Kauppila and colleagues[55] reported a computer-generated sequence and Villadsen and colleagues[56] used sequentially numbered sealed opaque envelopes. The staff administering the interventions were not blinded to allocation in any trial, and in three trials[37 55 57] outcomes were assessed by researchers aware of the treatment allocation. Using the ISPOR questionnaire, the quality of the model-based studies was generally good (figure 2C). However, none of the model-based studies reported a detailed process for internal and external validation. Four studies[42 44 50 52] were based on previously published models.[58–61] Of these, only Briggs and colleagues[58] (the basis for Fusco and

Turchetti's model[50]) provided details of model validation. Some model-based studies excluded potentially important outcomes, for example, Cummins and colleagues[46] did not include the possibility of requiring more than one revision surgery, and in Bolz and colleagues' model[53] there was no reduction in utility associated with delayed revisions. Further limitations in model validation were highlighted with the AdViSHE tool (figure 2D).[32]

The hierarchy of evidence used in the studies is reported in online supplementary table A6. Three model-based studies[42 44 46] scored poorly as they did not use randomised trial data to inform the clinical effective size of the interventions being analysed. Most studies did not cite a source for the assumption of the duration of treatment effect beyond that observed in the primary source for clinical effect size. Four trial-based studies and one model-based study included assessments of utilities from patients using validated tools and scored highly in terms of quality of evidence.

## Results of economic evaluation

The results of the included studies are summarised in table 2. Costs, QALYs and ICERs for each comparison performed in each study are listed in online supplementary table A7.

### Whole recovery pathway

Two studies considered multiple components of the enhanced recovery pathway. The first study was an economic analysis of a randomised trial with 56 THA and 31 TKA participants in Denmark.[38] The pathway used in the treatment arm ('accelerated care') had no difference from the control group ('conventional rehabilitation') in terms of intraoperative management, analgesia, nausea control or bowel regulation. Differences in the treatment protocols between the two arms involved patient education, nutrition, admission times, staffing and mobilisation (described in online supplementary table A8). The accelerated care pathway was the dominant strategy both overall and in the subgroup of THA patients (ie, less costly and more effective than the control group). For TKA patients, the authors found the accelerated care pathway to be cost-saving but less effective compared with the control group although not statistically significant. There was a cost saving of 618 075 Danish krone per QALY lost with accelerated care compared with conventional rehabilitation, which made it cost-effective (threshold of 160 000 krone per QALY in Denmark).

The second study was an economic evaluation of a before and after trial with 98 THA and 62 TKA participants in the Netherlands.[37] The intervention ('Joint Recovery Programme') consisted of a 20 min pre-assessment screening 6 weeks before the operation for physical assessment and analysis of the home situation to aid discharge planning, patient education sessions 1–2 weeks before surgery, group rehabilitation sessions and supervision by physical therapists and nurses (online supplementary table A8). Patients in the 'usual care' group underwent

**Table 1** Summary of studies included in this analysis

| Authors, publication year | Comparison | Joint | Study type | Country |
|---|---|---|---|---|
| **Enhanced recovery pathway** | | | | |
| Brunenberg et al, 2005[37] | Joint Recovery Programme (pre-assessment and intensive rehabilitation), vs conventional care | Hip and knee | Trial-based | The Netherlands |
| Larsen et al, 2009[38] | Accelerated perioperative care and rehabilitation, vs conventional care | Hip and knee | Trial-based | Denmark |
| **Preoperative** | | | | |
| McLawhorn et al, 2016[39] | Bariatric surgery followed by TKA 2 years later, vs immediate TKA | Knee | Markov model | USA |
| Fernandes et al, 2017[40] | Supervised neuromuscular exercise and an educational package, vs educational package alone | Hip and knee | Trial-based | Denmark |
| Courville et al, 2012[41] | Preoperative nasal screening for *Staphylococcus aureus* colonisation followed by mupirocin treatment for patients with positive cultures, vs empirical treatment of all preoperative patients with mupirocin, vs standard infection prevention measures without *S. aureus* screening or mupirocin decolonisation | Hip and knee | Decision tree model | USA |
| **Intraoperative** | | | | |
| Jackson et al, 2000[42] | Postoperative erythrocyte recovery and transfusion, vs usual transfusion practice | Hip and knee | Markov model | USA |
| Ramkumar et al, 2018[43] | Single-dose intravenous tranexamic acid, vs single-dose intravenous aminocaproic acid, vs no pharmacologic haemostatic agent | Hip and knee | Decision-tree model | USA |
| Sonnenberg, 2002[44] | Autologous blood donation and transfusion, vs usual practice without autologous donation | Hip | Markov model | USA |
| Marques et al, 2015[45] | Intraoperative local anaesthetic wound infiltration administered before wound closure in addition to standard anaesthesia, vs standard anaesthesia | Hip and knee | Trial-based | UK |
| Cummins et al, 2009[46] | Antibiotic-impregnated bone cement, vs conventional cement | Hip | Markov model | USA |
| Graves et al, 2016[47] | Nine arms, comparing combinations of prophylactic systemic antibiotics, antibiotic-impregnated cement, laminar airflow and body exhaust suits | Hip | Markov model | UK |
| Merollini et al, 2013[48] | No antibiotic prophylaxis, antibiotic prophylaxis and antibiotic-impregnated cement and antibiotic prophylaxis and laminar airflow, each compared with a baseline strategy of routine antibiotic prophylaxis | Hip | Markov model | Australia |
| Nherera et al, 2017[49] | Single-use negative pressure wound therapy dressings, vs usual care | Hip and knee | Decision-tree model | UK |
| **Postoperative** | | | | |
| Fusco and Turchetti, 2016[50] | 10 face-to-face rehabilitation sessions plus 10 telesessions, vs 20 face-to-face rehabilitation sessions | Knee | Markov model | Italy |
| Kauppila et al, 2011[51] | Multidisciplinary biopsychosocial outpatient rehabilitation programme, vs conventional orthopaedic care | Knee | Trial-based | Finland |
| Smith et al, 2018[52] | Telephonic health coaching and financial incentives vs telephone calls conveying general health messages | Knee | Markov model | USA |
| Bolz et al, 2010[53] | 2-yearly routine follow-up vs follow-up at 3 months and 1 or 2 years, vs no follow-up | Hip | Markov model | Australia |

Continued

## Table 1 Continued

| Authors, publication year | Comparison | Joint | Study type | Country |
|---|---|---|---|---|

TKA, total knee arthroplasty.

conventional physiotherapy for 1 hour/day and did not receive pre-assessment screening or information sessions, and discharge arrangements were addressed during admission to hospital. The joint recovery programme intervention was dominant for both hip and knee replacement, resulting in a cost saving of US$1261 per patient for THA and US$3336 per patient for TKA, with no statistically significant difference in effect. The probability that the joint recovery programme was the most cost-effective option was above 80% for THA and TKA for willingness to pay thresholds up to US$45 000.

### Preoperative components

McLawhorn and colleagues[39] used a Markov model to assess the cost-effectiveness of bariatric surgery 2 years before TKA for morbidly obese (body mass index≥35 kg/m$^2$) patients who were candidates for both operations due to end-stage knee osteoarthritis and failed non-operative weight-loss interventions. The strategy including bariatric surgery was cost-effective at the stated willingness to pay threshold of US$100 000 per QALY in 98.8% of probabilistic simulations.

Fernandes and colleagues[40] conducted an economic evaluation alongside a trial of 8 weeks of supervised neuromuscular exercise in addition to an educational package prior to surgery. Their point estimate was that the intervention was dominant, with a saving of 132 euro and a benefit of 0.04 QALYs. It had an 84% probability of being cost-effective at a willingness to pay threshold of 40 000 euro per QALY.

Courville and colleagues[41] compared three strategies of screening for and treating *Staphylococcus aureus* colonisation to prevent deep surgical site infections for TKA and THA. They found that decolonisation of all preoperative patients with mupirocin, without testing for *S. aureus*, was the dominant strategy (cheaper and more effective) when compared with treating patients testing positive only for *S. aureus*, or no screening or decolonisation for *S. aureus*.

### Intraoperative components

Strategies to reduce allogenic blood transfusions were collection of autologous blood prior to surgery,[44] aseptic collection of wound drainage[42] and use of aminocaproic acid or tranexamic acid to reduce bleeding.[43] Autologous blood collection was found to be cost-effective at US$2750 per QALY gained, whereas wound drainage collection was not, costing US$5.7 million per QALY gained. However, in the latter study, the only benefit of avoiding allogenic transfusion considered was a reduced risk of blood-borne virus infection.[42] In contrast, Sonnenberg's model for THA[44] found a minimal effect from the risk of blood-borne virus infection in either costs or outcomes: 99.6% of the increase in QALYs was due to a reduced risk of bacterial infection. When the risk of bacterial infection was removed from this model, the ICER increased to US$2.5 million per QALY gained. Use of tranexamic acid was more effective and cost-saving compared with either aminocaproic acid or not using a haemostatic agent.[43]

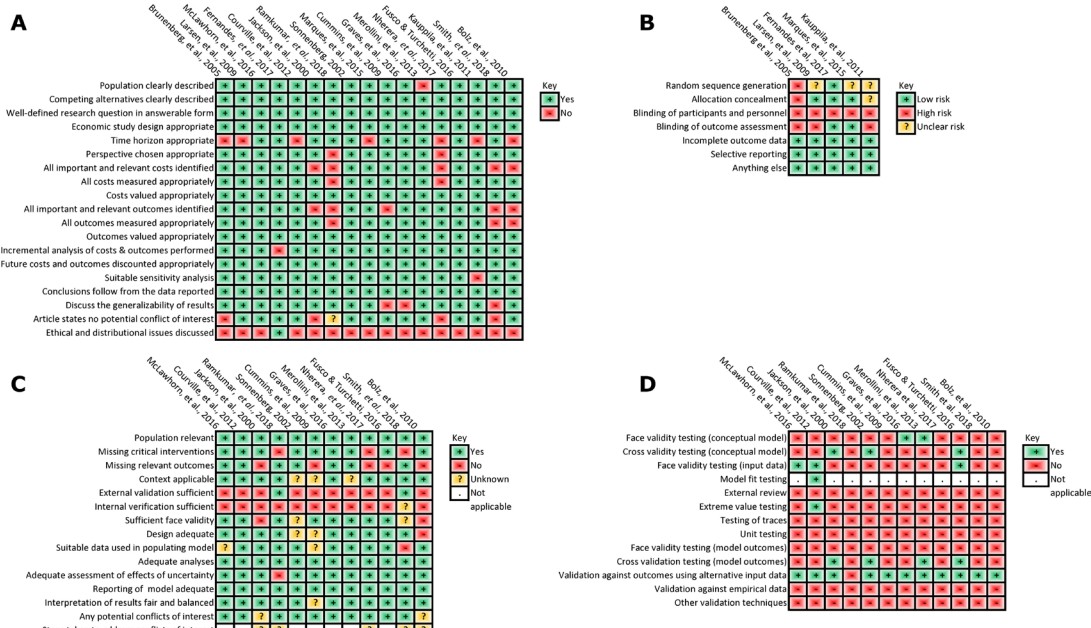

**Figure 2** Assessments of study quality based on tools from (A) Consensus on Health Economic Criteria,[30] (B) International Society for Pharmacoeconomics and Outcomes Research,[31] (C) assessment of the validation status of Health-Economic decision models tool[32] and (D) Cochrane Collaboration.[33] Note that the study by Brunenberg and colleagues[37] was a non-randomised before and after trial and we have included it in panel (D) for completeness.

Marques and colleagues conducted economic evaluations[45] alongside two randomised controlled trials of adding local wound infiltration with bupivacaine to usual anaesthetic care for THA and TKA.[57] The infiltration of local anaesthetic was found to be dominant compared with standard anaesthesia in both THA and TKA patients.

Three studies used Markov models to investigate similar measures to reduce surgical site infection in the USA,[46] the UK[47] and Australia.[48] The dominant strategy in all the three included the use of antibiotic-impregnated cement. The two studies that looked at other factors[47 48] each found use of prophylactic systemic antibiotics to be dominant over non-use, and use of conventional ventilation in operating theatres to be dominant over laminar airflow ventilation. Graves and colleagues[47] considered the use of body exhaust suits and found them to be dominated by strategies that did not include use of these suits. A fourth study investigated using single-use negative pressure wound dressings to reduce surgical site infection.[49] This was found to be dominant compared with usual care.

### Postoperative components

Fusco and Turchetti[50] used a Markov model to evaluate a strategy of 10 face-to-face rehabilitation sessions followed by 10 telerehabilitation sessions after TKA, compared with 20 face-to-face sessions. They found the strategy including telerehabilitation to be cost saving, and improved range of movement (knee flexion). However, they found no utility data for patients following a telerehabilitation programme so for their base case assumed it to be non-inferior to face-to-face rehabilitation. In a sensitivity analysis, if telerehabilitation conferred an improvement in quality of life of at least 2.5%, the strategy's probability

of being cost-effective was 1; if it led to a reduction in quality of life of at least 2.5%, the probability of being cost-effective was 0 (each at a willingness to pay threshold of 30 000 euros per QALY).

Kauppila and colleagues[51] performed an economic evaluation of a 10-day outpatient rehabilitation course between 2 and 4 months after TKA which included clinical assessments, physical activity, sessions with a psychologist and lectures from an orthopaedic surgeon, nutritionist and social worker.[55] They found that patients who completed this course had higher costs and slightly worse quality of life outcomes over 1 year of follow-up (though the difference was non-significant at the 5% level) compared with those receiving conventional orthopaedic care.

Smith and colleagues[52] used a Markov model to investigate a postoperative strategy combining telephonic health coaching and financial incentives to increase physical activity. The intervention increased costs by US$300 and was associated with an increase in utility of 0.005 QALYs. The point-estimate ICER was US$57 200 per QALY, with a 70% probability of being cost-effective at a willingness to pay threshold of US$100 000 per QALY.

Bolz and colleagues[53] compared three follow-up strategies: two yearly routine follow-up; follow-up twice (at 3 months, and between 1 and 2 years after surgery) or no follow-up. The model assumed that no revisions would be delayed in either strategy that included follow-up, and the outcomes for these two strategies were identical in each analysis. The no follow-up strategy was dominant for any assumed rate of delayed revision between 1% and 50%.

**Table 2** Summary of findings from studies included in this analysis

| Authors, country | Population | Strategy | Cost-effective? |
|---|---|---|---|
| **Enhanced recovery pathway** | | | |
| Brunenberg et al, The Netherlands[37] | THA and TKA | Conventional care | – |
| | | Joint Recovery Programme (pre-assessment and intensive rehabilitation) | Yes, more effective and less costly |
| Larsen et al, Denmark[38] | THA | Conventional care | – |
| | | Accelerated perioperative care and rehabilitation | Yes, more effective and less costly |
| | TKA | Conventional care | – |
| | | Accelerated perioperative care and rehabilitation | Yes, less effective but less costly |
| **Preoperative** | | | |
| McLawhorn et al, USA[39] | Morbid obese TKA | Immediate TKA | – |
| | | Bariatric surgery, followed by TKA 2 years later | Yes |
| Fernandes et al, Denmark[40] | THA and TKA | Educational package | – |
| | | Supervised neuromuscular exercise in addition to educational package | Yes, more effective and less costly |
| Courville et al, USA[41] | THA and TKA | Standard infection prevention measures without *Staphylococcus aureus* screening or mupirocin decolonisation OR preoperative nasal screening for *S. aureus* followed by mupirocin treatment for patients with positive cultures | – |
| | | Empirical treatment of all preoperative patients with mupirocin | Yes, more effective and less costly |
| **Intraoperative** | | | |
| Jackson et al, USA[42] | THA and TKA | Usual transfusion practice | – |
| | | Postoperative erythrocyte recovery and transfusion | No |
| Ramkumar et al, USA[43] | THA and TKA | No pharmacologic haemostatic agent OR single-dose intravenous aminocaproic acid | – |
| | | Single-dose intravenous tranexamic acid | Yes, more effective and less costly |
| Sonnenberg, USA[44] | THA | Usual practice without autologous donation | – |
| | | Autologous blood donation and transfusion | Yes |
| Marques et al, UK[45] | THA and TKA | Standard anaesthesia | – |
| | | Intraoperative local anaesthetic wound infiltration administered before wound closure in addition to standard anaesthesia | Yes, more effective and less costly |
| Cummins et al, USA[46] | THA | Conventional cement | – |
| | | Antibiotic-impregnated bone cement | Yes, more effective and less costly |
| Graves et al, UK[§47] | THA | No systemic antibiotics, plain cement and conventional ventilation | – |
| | | Systemic antibiotics, antibiotic-impregnated cement and conventional ventilation | Yes, more effective and less costly |
| Merollini et al, Australia[48] | THA | No antibiotic prophylaxis OR antibiotic prophylaxis OR antibiotic prophylaxis and laminar airflow | – |
| | | Antibiotic prophylaxis and antibiotic-impregnated cement | Yes, more effective and less costly |
| Nherera et al, UK[49] | THA and TKA | Usual care | – |
| | | Single-use negative pressure wound therapy dressings | Yes, more effective and less costly |
| **Postoperative** | | | |
| Fusco and Turchetti, Italy[50] | TKA | 20 face-to-face rehabilitation sessions | – |
| | | 10 face-to-face rehabilitation sessions plus 10 telesessions | Yes, same effectiveness but less costly |

**Table 2** Continued

| Authors, country | Population | Strategy | Cost-effective? |
|---|---|---|---|
| Kauppila et al, Finland[51] | TKA | Conventional orthopaedic care | – |
| | | Multidisciplinary biopsychosocial outpatient rehabilitation programme | No |
| Smith et al, USA[52] | TKA | Telephone calls conveying general health messages | – |
| | | Telephonic health coaching and financial incentives to increase physical activity | Yes |
| Bolz et al, Australia[53] | THA | 2-yearly routine follow-up OR follow-up at 3 months and 1 or 2 years | – |
| | | No follow-up | Yes, more effective and less costly |

THA, total hip arthroplasty; TKA, total knee arthroplasty.

## Effects of standardising currencies and price years

Most studies showed one strategy to be dominant (ie, cheaper and more effective) over the others[37 38 40 41 43 45–51 53] and were therefore not affected by changes in currency or price year. When converted into 2016 US PPP, willingness to pay thresholds ranged between US$22 112 (Denmark) and US$100 000 (USA, online supplementary table A9). Autologous blood transfusion[44] and bariatric surgery[39] would be cost-effective across all willingness to pay thresholds identified in the review. The capture and replacement of red cells[42] cost 7.8 million US PPP dollars per QALY gained and would not be cost-effective by any study's thresholds. Telephonic health coaching and financial incentives to increase physical activity[52] were cost-effective at a willingness to pay of US$100 000 per QALY but not with any lower threshold. Accelerated care pathway in TKA patients in Denmark[38] would not be cost-effective in a US setting using a willingness to pay of US$100 000 per QALY.

## DISCUSSION

### Summary of evidence

Our objective was to assess the cost-effectiveness evidence of enhanced recovery following THA or TKA through a systematic collection of published cost–utility data. Previous systematic reviews considered effectiveness[16] and patient satisfaction,[62] but we believe that this is the first systematic review assessing the cost-effectiveness of enhanced recovery for THA or TKA patients. We identified two cost–utility analyses of an entire pathway. This is consistent with reviews of cost-effectiveness of enhanced recovery programmes for other surgical sites,[63–65] which have found few studies that reported the effect on quality of life and none presented the cost-effectiveness results using QALYs. Both studies that considered cost-effectiveness of an entire recovery pathway were trials in both TKA and THA patients.[37 38] The enhanced recovery pathway was found to be associated with reduced costs for all patients and the incremental cost-effectiveness estimate favoured the enhanced recovery protocol, with a high probability of being the most cost-effective option.

We identified 15 studies presenting cost–utility data for components of an enhanced recovery pathway. These studies covered only a few of the potential enhanced recovery pathway components and were conducted across different healthcare systems using different cost perspectives. Three studies investigated overlapping strategies for reducing surgical infections supporting the use of prophylactic systemic antibiotics, antibiotic-impregnated cement and conventional ventilation.[46–48] No other interventions were examined by more than one study. Scope to combine or generalise results is therefore limited.

The studies identified in this review were generally of good quality according to the CHEC list[30] with a short time horizon identified as a key limitation in nine studies (between 6 weeks and 7 years). This is of significant concern as short time horizons will not capture or model the impact of the interventions on costs and benefits accruing over the long post-acute care period of interest. Furthermore, we also found that the large majority of studies did not consider ethical aspects and distributional implications of their findings. When the models were assessed against the ISPOR questionnaire,[31] there were concerns about the lack of model validation work, potentially questioning the reliability of 10 of the 12 studies identified. The trials were generally of good quality. One trial was a non-randomised before and after trial,[37] and participants were not blinded to the intervention for postoperative interventions in any of the studies. However, we concluded that the overall risk of bias among the studies appeared low.

### Limitations of this review

We may have missed relevant evidence by limiting our search to reports published in the English language and excluding studies that did not report QALYs. Cost–utility analyses enable comparison between different interventions and are the preferred intelligence for healthcare allocation decisions.[18 19] We therefore felt justified in limiting our review to studies reporting QALYs. We identified one study excluded due to not reporting QALYs that investigated a complete enhanced recovery pathway.[66] In this study, the recovery pathway was found to be cost-saving and associated with statistically significant differences in

knee flexion and extension at 6 months. However, the clinical significance of these differences was not discussed and no health-related quality of life data were collected.[66]

The second limitation is that standard practices have changed during the period covered by the included studies. For example, many enhanced recovery techniques are now standard practice,[8] and Nherera and colleagues[49] even included enhanced recovery pathway as usual care in their model. Furthermore, other practices in the included studies are now outdated. For example, patients in the control group of Brunenberg and colleagues' trial had had an in-patient stay of 9.4 days,[37] whereas by 2015 the mean length of stay in England was 4.9 days for TKA and 5.4 days for THA[8] and has further decreased since.[67] Also, practices such as blood transfusion rates have changed greatly: the two studies included in this review assumed transfusion rates of 89%[44] to 100%,[42] whereas rates in 2014 were 22% for THA and 18% for TKA patients,[68] and rates of infection from blood transfusion are now lower than those used in the models.[69]

## Research gaps for future work

Use of prophylactic systemic antibiotics and antibiotic-impregnated cement, and thromboprophylaxis[23–25] are the only measures investigated in more than one cost–utility analysis. There is therefore very wide scope for further investigation of components of enhanced recovery. Pathways have a 'whole package' benefit in excess of the sum of their parts and can be used to ensure dissemination of evidence-based practice and delivery of more consistent care.[70–72] From this perspective, specific components within a pathway might be less important than the degree of compliance with a protocol. However, as the use of care pathways becomes more prevalent, optimising their components will be the next step in improving outcomes.

## CONCLUSIONS

There is limited cost–utility evidence, either for an entire enhanced recovery pathway or for individual components of a pathway, for patients having THA or TKA. There are also concerns regarding the ability of short time horizons in trials in this area to capture relevant outcomes, and regarding a general lack of reporting of model validation. Our findings support the use of enhanced recovery pathways as a whole, prophylactic systemic antibiotics, antibiotic-impregnated cement and conventional ventilation. No other interventions were assessed by more than one study. Those single studies supported use of empirical preoperative *S. aureus* decolonisation of all patients with mupiricin, single-dose intravenous tranexamic acid, wound infiltration with local anaesthetic, single-use negative pressure wound therapy dressings, bariatric surgery for morbidly obese patients requiring TKA, use of telerehabilitation and telephonic health coaching with financial incentives to increase physical activity. However, we were unable to conclude how transferable these findings

would be into other healthcare systems. There is ample scope for future cost-effectiveness studies into enhanced recovery for THA and TKA patients. In particular, we recommend analysis of entire enhanced recovery pathways and comparison of incremental changes of components within pathways rather than considering aspects of care in isolation.

**Author affiliations**
[1]Nuffield Department of Population Health, University of Oxford, Oxford, UK
[2]John Radcliffe Hospital, Oxford University Hospitals NHS Foundation Trust, Oxford, UK
[3]Wolfson Institute of Preventive Medicine - Barts and the London, Queen Mary University of London, London, UK
[4]Stoke Mandeville Hospital, Buckinghamshire Healthcare NHS Trust, Aylesbury, UK
[5]Musculoskeletal Research Unit, Translational Health Sciences, Bristol Medical School, University of Bristol, Bristol, UK
[6]Nuffield Department of Orthopaedics, Rheumatology and Musculoskeletal Sciences (NDORMS), University of Oxford, Oxford, UK

**Acknowledgements** We are grateful to Nia Roberts, information specialist, and Rafael Pinedo-Villanueva, senior research associate in health economics, both at the University of Oxford, for their assistance with the search strategy.

**Contributors** JM, AJ and JL designed the study and all authors defined the inclusion criteria. JM, LC and JL developed the search strategies. JM and LC piloted the search strategy and inclusion criteria. JM and JL developed the data extraction pro forma. JM, LC and JL screened the studies. JM, MGP and RJ extracted the data. MGP wrote the draft of the manuscript. JM and JL provided critical revisions to the article and all authors approved the final version of the article to be published. JL is the guarantor of the review.

**Funding** This project was funded by the National Institute for Health Research Health Services and Delivery Research Programme (project number 14/46/02). Support was received from the Oxford NIHR Biomedical Research Centre, Nuffield Orthopaedic Centre, University of Oxford. AJ was supported by the NIHR Biomedical Research Centre at the University Hospitals Bristol NHS Foundation Trust and the University of Bristol.

**Disclaimer** The views and opinions expressed herein are those of the authors and do not necessarily reflect those of the HS&DR Programme, NIHR, NHS or the Department of Health and Social Care.

**ORCID iDs**
Mark G Pritchard http://orcid.org/0000-0003-1726-8989
Jacqueline Murphy http://orcid.org/0000-0002-3927-7002
Andrew Judge https://orcid.org/0000-0003-3015-0432
Jose Leal http://orcid.org/0000-0001-7870-6730

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
