## [Reviewer comments · BMJ Open]

ARTICLE DETAILS

TITLE (PROVISIONAL)	Enhanced recovery following hip and knee arthroplasty: A systematic review of cost-effectiveness evidence
AUTHORS	Pritchard, Mark; Murphy, Jacqueline; Cheng, Lok; Janarthanan, Roshni; Judge, Andrew; Leal, Jose

VERSION 1 - REVIEW

REVIEWER	Ton Lenssen Maastricht University Medical center, Netherlands
REVIEW RETURNED	01-Jul-2019

GENERAL COMMENTS	I think the manuscript is very well written and the study procedures are state of the art. Yet my main concern, which is shared by the authors in the discussion section, focuses on the enormous change in practice over the last decennium. Therewith in my opinion the information in this review is for a part out-dated. The study of Brunenberg et al (2000) for example compares a 'new' procedure to a care model in which patients were hospitalised approximately 10 days, which in my opinion is not state of the art in 2019. The information that the programme was cost effective is most certainly true but I question the actuality en therewith current relevance. Therefore I would suggest to make revisions to the manuscript by adding clinical relevance to the in- and exclusion criteria of studies and therewith exclude studies describing obsolete or meanwhile globally accepted standards of care. Minor comments Page 6 line 2: I was surprised that pre-operative improvement of physical functioning wasn't included in box 1 since it is a hot topic in preoperative management nowadays. Page 6 lines 40 and 41: Why did the authors include studies with patients with other indications than osteoarthritis since this may well lead to selection bias especially when independent extraction of data for the OA population wasn't possible. The fact that overall indication is over 90% OA does not justify the choice since it is not reported how the distribution was in the individual studies. Page 11 it is confusing that authors us references out of the main text referring to a table in which references are used for the same articles but with references from the secondary reference list Page 14 lines 1-22: it remains unclear what treatment Larsen's conventional care group received, so the contrast between treatments remains unknown Page 14 lines 22- In my opinion data of this study is out-dated.
---

	Page 16 lines 40-51: I understand receiving extra attention without economic benefit or improved quality of life is evidently worse. I lack information on the reasons for including patients in the ten day course, which leaves me with a blind spot of what authors thought were of providing patients with a programme in the first place. Page 19 line.. a second limitation ... studies I fully agree with the authors and therefore suggest adding clinical relevance to the in- and exclusion criteria of studies and therewith exclude studies describing obsolete or meanwhile globally accepted standards of care.
--	---

REVIEWER	William B Weeks, MD, PhD, MBA Microsoft Healthcare NExT USA
REVIEW RETURNED	16-Jul-2019

GENERAL COMMENTS	This is a well-written paper on an important topic. It is clear, concise, and nicely summarizes the findings of the review. The only suggestions I have are 1. to include, in the concerns, the lack of ethical considerations (noted in Figure 2) that apparently applied to all studies and 2. to reiterate that these studies focus on inpatient care (and short term follow up) and true cost-utility analyses of costs should include the 'bundle' cost - there is a literature showing considerable variation in post-acute care costs (which represent most of the care costs) - optimization of post-acute care pathways might be as important as optimization of index hospitalization for TKA and THA.
---

VERSION 1 – AUTHOR RESPONSE

#Reviewer(s)' Comments to Author:

Reviewer: 1

Reviewer Name: Ton Lenssen

Institution and Country: Maastricht University Medical center, Netherlands Please state any competing interests or state 'None declared': None declared

I think the manuscript is very well written and the study procedures are state of the art. Yet my main concern, which is shared by the authors in the discussion section, focuses on the enormous change in practice over the last decennium. Therewith in my opinion the information in this review is for a part out-dated.

The study of Brunenberg et al (2000) for example compares a 'new' procedure to a care model in which patients were hospitalised approximately 10 days, which in my opinion is not state of the art in 2019. The information that the programme was cost effective is most certainly true but I question the actuality en therewith current relevance.

Therefore I would suggest to make revisions to the manuscript by adding clinical relevance to the in- and exclusion criteria of studies and therewith exclude studies describing obsolete or meanwhile globally accepted standards of care.

AUTHORS' REPLY: We thank the reviewer for the positive feedback. We agree that clinical practice has changed in many countries in recent years with considerable reduction in the length of stay associated with joint replacement. However, some of the more historical comparisons still provide useful data as they capture the additional benefit and costs of interventions before they became part of usual care. We have added the following to the discussion:

'A second limitation is that standard practices have changed during the period covered by the included studies. For example, many enhanced recovery techniques are now standard practice, and Nherera and colleagues even included an enhanced recovery pathway as usual care in their model. Furthermore, other practices in the included studies are now outdated. For example, patients in the control group of Brunenberg and colleagues' trial had an inpatient stay of 9.4 days whereas by 2015, the mean length of stay in England was 4.9 days for TKA and 5.4 days for THA and has further decreased since. Also, practices such as blood transfusion rates have changed greatly: the two studies included in this review assumed transfusion rates of 89% to 100%, whereas rates in 2014 were 22% for THA and 18% for TKA patients, and rates of infection from blood transfusion are now lower than those used in the models.'

Minor comments

Page 6 line 2: I was surprised that pre-operative improvement of physical functioning wasn't included in box 1 since it is a hot topic in preoperative management nowadays.

AUTHORS' REPLY: We have revised Box 1 to better reflect components of enhance recovery pathway by adding items such as neuromuscular electrical stimulation, nutrition screening, physiotherapy, pulsed electromagnetic fields, computer-assisted surgery, and wound care.

Page 6 lines 40 and 41: Why did the authors include studies with patients with other indications than osteoarthritis since this may well lead to selection bias especially when independent extraction of data for the OA population wasn't possible. The fact that overall indication is over 90% OA does not justify the choice since it is not reported how the distribution was in the individual studies.

AUTHORS' REPLY: We understand the point the reviewer is making. However, this was a pragmatic choice when writing the protocol and the large majority of identified studies consist solely of OA populations. The weight of <10% of non OA population in the cost-utility analysis is very unlikely to change the findings of the studies.

Page 11 it is confusing that authors us references out of the main text referring to a table in which references are used for the same articles but with references from the secondary reference list

AUTHORS' REPLY: We have standardised the references in manuscript and online appendix.

Page 14 lines 1-22: it remains unclear what treatment Larsen's conventional care group received, so the contrast between treatments remains unknown Page 14 lines 22- In my opinion data of this study is out-dated.

AUTHORS' REPLY: Details of Larsen's conventional care group are given in Supplementary Table A8, which lists differences including in pre-operative education, scheduling of admission, nutrition and mobilization.

Page 16 lines 40-51: I understand receiving extra attention without economic benefit or improved quality of life is evidently worse. I lack information on the reasons for including patients in the ten day course, which leaves me with a blind spot of what authors thought were of providing patients with a programme in the first place.

AUTHORS' REPLY: More information on Kappila et al. study is reported in Table 1 and supplementary Tables A3 and A7. The population consisted of 60-80 year olds having unilateral TKR for knee osteoarthritis. The authors' hypothesis was that multidisciplinary rehabilitation program in outpatient care setting would yield faster attainment of functional recovery than would conventional orthopaedic care 1 year after surgery. Conventional care included physiotherapy. This consisted of preoperative guided exercises, a daily guided exercise program on the surgical ward, and a guided subsequent exercise program at a 2-month outpatient control visit to an orthopedic surgeon. Finally, according to the authors, patients in Finland are predominantly treated in the acute hospital system with greater use of outpatient physiotherapy than inpatient, multidisciplinary rehabilitation with extended-care facilities, including occupational therapy.

Page 19 line.. a second limitation ... studies I fully agree with the authors and therefore suggest adding clinical relevance to the in- and exclusion criteria of studies and therewith exclude studies describing obsolete or meanwhile globally accepted standards of care.

AUTHORS' REPLY: We thank the reviewer for this comment. As mentioned above we added the following to the discussion:

'A second limitation is that standard practices have changed during the period covered by the included studies. For example, many enhanced recovery techniques are now standard practice, and Nherera and colleagues even included an enhanced recovery pathway as usual care in their model. Furthermore, other practices in the included studies are now outdated. For example, patients in the control group of Brunenberg and colleagues' trial had an inpatient stay of 9.4 days whereas by 2015, the mean length of stay in England was 4.9 days for TKA and 5.4 days for THA and has further decreased since. Also, practices such as blood transfusion rates have changed greatly: the two studies included in this review assumed transfusion rates of 89% to 100%, whereas rates in 2014 were 22% for THA and 18% for TKA patients, and rates of infection from blood transfusion are now lower than those used in the models.'

Reviewer: 2

Reviewer Name: William B Weeks, MD, PhD, MBA

Institution and Country: Microsoft Healthcare NExT, USA

Please state any competing interests or state 'None declared': None declared

This is a well-written paper on an important topic. It is clear, concise, and nicely summarizes the findings of the review.

The only suggestions I have are 1. to include, in the concerns, the lack of ethical considerations (noted in Figure 2) that apparently applied to all studies and 2. to reiterate that these studies focus on inpatient care (and short term follow up) and true cost-utility analyses of costs should include the 'bundle' cost - there is a literature showing considerable variation in post-acute care costs (which represent most of the care costs) - optimization of post-acute care pathways might be as important as optimization of index hospitalization for TKA and THA.

AUTHORS' REPLY: We thank the reviewer for the positive feedback. We have added the following to the discussion:

“The studies identified in this review were generally of good quality according to the CHEC list with a short time horizon identified as a key limitation in nine studies (between 6 weeks and 7 years). This is of significant concern as short time horizons will not capture or model the impact of the interventions on costs and benefits accruing over the long post-acute care period of interest. Furthermore, we also found that the large majority of studies did not consider ethical aspects and distributional implications of their findings.”

VERSION 2 – REVIEW

REVIEWER	Ton Lenssen MUMC+, Maastricht, Netherlands
REVIEW RETURNED	19-Nov-2019

GENERAL COMMENTS	My main concern in the prior submission was the enormous change in practice over the last two decades, leading to the fact that some studies yielding positive results are outdated at this point in time. Authors address this in this version in their discussion section. Since the study is conducted in a state of the art way, the only remaining discussion for me would be whether the informativity of the given information is at part not outdated and therefore less interesting for the reader. Since I have no further comments on the manuscript itself I feel that the manuscript can be published. I think the editors must choose whether they want to publish systematic review in which partly outdated conclusions are incorporated.
---